# Personalized Medicine Workflow in Post-Traumatic Orbital Reconstruction

**DOI:** 10.3390/jpm12091366

**Published:** 2022-08-24

**Authors:** Juliana F. Sabelis, Ruud Schreurs, Harald Essig, Alfred G. Becking, Leander Dubois

**Affiliations:** 1Department of Oral and Maxillofacial Surgery, Amsterdam University Medical Centre (UMC), AMC, Academic Center for Dentistry Amsterdam (ACTA), Meibergdreef 9, 1105 AZ Amsterdam, The Netherlands; 2Department of Oral and Maxillofacial Surgery, Radboud University Medical Centre Nijmegen, Geert Grooteplein Zuid 10, 6525 GA Nijmegen, The Netherlands; 3Department Oral & Maxillofacial Surgery, University Hospital, Frauenklinikstrasse 24, 8091 Zurich, Switzerland

**Keywords:** patient-specific implants, orbital reconstruction, computer-assisted surgery, surgical navigation, additive manufacturing

## Abstract

Restoration of the orbit is the first and most predictable step in the surgical treatment of orbital fractures. Orbital reconstruction is keyhole surgery performed in a confined space. A technology-supported workflow called computer-assisted surgery (CAS) has become the standard for complex orbital traumatology in many hospitals. CAS technology has catalyzed the incorporation of personalized medicine in orbital reconstruction. The complete workflow consists of diagnostics, planning, surgery and evaluation. Advanced diagnostics and virtual surgical planning are techniques utilized in the preoperative phase to optimally prepare for surgery and adapt the treatment to the patient. Further personalization of the treatment is possible if reconstruction is performed with a patient-specific implant and several design options are available to tailor the implant to individual needs. Intraoperatively, visual appraisal is used to assess the obtained implant position. Surgical navigation, intraoperative imaging, and specific PSI design options are able to enhance feedback in the CAS workflow. Evaluation of the surgical result can be performed both qualitatively and quantitatively. Throughout the entire workflow, the concepts of CAS and personalized medicine are intertwined. A combination of the techniques may be applied in order to achieve the most optimal clinical outcome. The goal of this article is to provide a complete overview of the workflow for post-traumatic orbital reconstruction, with an in-depth description of the available personalization and CAS options.

## 1. Introduction

The orbit is an inward-projecting bony structure in the shape of a cone (or pyramid) at the transition between midface and skull base [1,2,3]. The base of the orbit, the orbital rim, is composed of thick bone; in contrast, the orbit’s inner walls are thin bony structures. The orbit provides the casing for the soft-tissue structures associated with the visual (motor) system: neurovascular structures, connective tissue, ocular muscles, and the globe [4,5].

With its central position and thin bony walls, the orbit is probably the most vulnerable part of the facial skeleton [1,2]. Two possible theories of orbital fracture pathogenesis have been suggested. The buckling theory suggests that the energy of a traumatic impact on the orbital rim after blunt-force trauma is propagated to the thin inner walls and leads to a fracture in these weaker structures [6,7,8]. Hydraulic theory considers increased pressure after impact on the globe and orbital contents as the main reason for orbital wall fractures. The exact nature and specifics of a fracture may be explained by a combination of both mechanisms [7].

The orbital volume may be increased, and soft tissue may be displaced into the adjacent sinuses due to the impact or the dislocation of supporting bony structures. The globe’s position may be displaced after trauma, for instance, with inward displacement (enophthalmos) or inferior displacement (hypoglobus). The orbital soft tissue may be affected by the traumatic impact as well. The structural integrity and functional capacities of connective tissue or extraocular muscles may be disrupted, resulting in a disturbance of ocular motility and double vision (diplopia). The location and type of the impact, in combination with the amount of energy transferred to the orbit’s bony structures and orbital soft tissue, are responsible for the heterogeneity in clinical presentation.

There is an ongoing debate on the indication of surgical reconstruction, and systematic reviews have not been able to provide evidence-supported guidelines [9,10,11]. Some advocate a radical approach to prevent clinical symptoms [12], while others choose a more conservative approach with a delayed surgery if clinical symptoms develop [13]. Indication for reconstruction remains a subjective decision in most cases, depending on the surgeon and patient characteristics. The surgical management of orbital fractures focuses on the repositioning the orbital contents and the globe and reinstating the structural support to recover ocular function. Restoration of the orbit is the first and most predictable step in the surgical treatment of orbital fractures [14,15].

Nowadays, titanium mesh implants have become the preferred biomaterial for the surgical reconstruction of the orbit. Titanium implants can be categorized into flat implants, preformed implants, and patient-specific implants (PSIs). Flat implants are manually shaped and trimmed by the surgeon. A generic or individual model of the (mirrored) orbit may aid in the molding process. Preformed implants have a predefined shape, based on a model of the average orbit [10,16]. Patient-specific implants (PSIs) are designed on an individual basis for the patient and are subsequently produced through additive manufacturing.

The soft tissue’s intricate architecture and the proximity to vital structures pose surgical challenges in orbital reconstruction [17,18]. Orbital reconstruction is keyhole surgery performed in a confined space. This contributes to limited visualization, which is further enhanced by protruding fat. The margin of error is small: an incorrectly positioned implant may have significant implications for the clinical outcome and the patient’s quality of life, and it is considered a ground for revision surgery in the literature [19,20]. Medical technology has been incorporated in the clinical workflow of orbital reconstructions to reduce the risk of implant malpositioning [21].

This technology-supported workflow, called computer-assisted surgery (CAS), has become the standard for complex orbital traumatology in many hospitals [22]. The introduction of CAS has also enabled personalization of the treatment: treatment planning is customized to fit the options and needs of the patient, and intraoperative guidance is adjusted to the anatomical possibilities. The main aim of this article is to provide a complete overview of the (CAS) workflow for orbital reconstruction, with an in-depth description of the techniques embedded in the workflow and with a special focus on treatment personalization through patient-specific implant design.

## 2. Post-Traumatic Orbital Reconstruction Workflow

The conventional workflow of post-traumatic orbital reconstruction and possible CAS techniques are illustrated in Figure 1. The individual phases are explained in detail in the following paragraphs.

### 2.1. Diagnostics

A thorough clinical and radiographic evaluation of the patient is essential to determine the optimal treatment. Clinical evaluation should at least assess the amount of globe displacement and the degree of double vision. The Hertel exophthalmometer is the simplest tool to quantitatively measure the relative ventrodorsal globe position [23]. It is the current gold standard despite the known limitations, such as the asymmetry of lateral orbital rims, the compression of soft tissue, and the lack of a uniform technique [23,24]. There are no readily available reproducible tools for measuring the relative craniocaudal globe position [25]. It is assessed by the Hirschberg test, which evaluates the light reflex centered on each pupil to reveal vertical asymmetry [26]. Alternative methods have been proposed to quantify globe position differences based on imaging, but these have not been broadly implemented [23,27].

One of the difficulties in clinical decision-making is to address the most common complaint in orbital fractures: diplopia. It is challenging to find the actual cause of diplopia; in most cases, it is caused by a restriction of ocular motility. Ocular motility can be disturbed by impingement or the entrapment of the ocular muscles and surrounding soft tissue, but also by muscle edema, muscle injury, hemorrhage, emphysema, or motor nerve palsy. In a trauma setting, orthoptic measurements may be challenging to perform due to logistics, limited mobility of the patient, or considerable periorbital swelling. Absolute restrictions, as seen in trapdoor fractures, need to be treated shortly after the trauma. Ocular motility deficits with different etiology may improve or resolve over time, and surgery might not be indicated. In these cases, it is advisable to perform several orthoptic assessments over time to monitor spontaneous improvement. Moreover, the orthoptist may be able to differentiate between possible causes of double vision through repeated measurements [9,28].

Computed tomography (CT) is the modality of choice for radiographic evaluation because of the superior visualization of bony structures. The size and extent of the fracture may be estimated or measured in the coronal, sagittal, or axial plane. Considering that the bone is paper-thin in certain areas, a maximum slice thickness of 1.0 mm is essential for evaluation. In individual cases, the evaluation of soft-tissue changes may become important. Shape alterations of the inferior rectus muscle have been reported to affect delayed or postoperative enophthalmos [29,30,31] and may affect treatment decisions [10]. In addition, herniation of the orbital soft tissue might be an indication for surgical reconstruction. Magnetic resonance imaging (MRI) provides superior soft-tissue contrast compared to CT and is more sensitive for identifying extraocular muscle or periorbital fat entrapment [32,33]. Nevertheless, the acquisition of an MRI is not part of the standard imaging protocol for orbital trauma [13]. This may change in the future, considering that all subsequent treatment steps benefit from optimizing the information collection in the diagnostics phase.

### 2.2. Advanced Diagnostics

Advanced diagnostics aim to maximize the information extracted from the available image data. For this purpose, the CT scan is imported into the virtual surgical planning software. The CT scan is subdivided into voxels (3D pixels), each with a grayscale value corresponding to the X-ray absorption within that volume. These voxels may be segmented (grouped) based on the tissue type or anatomical structure they belong to. Anatomical structures of interest in orbital trauma are the orbit, orbital cavity, and possibly surrounding bony structures such as the zygomatic complex. The segmentation is visualized as an overlay in the multi-planar view and as a 3D model. Additional information may be collected through quantification (e.g., volume measurement), or manipulation (e.g., mirroring) of the segmented anatomy (illustrated in Figure 2). The unaffected contralateral orbit and orbital cavity can provide a reference for the affected orbit in unilateral fractures, which provides insight into the extent of the fracture and displacement of orbital walls or surrounding bony structures. The volume of the affected orbit can be compared to the unaffected healthy side to determine the relative volume change, since it has been proven that the orbits are highly symmetrical [33]. These volumetric changes can be incorporated into the treatment plan [34]. Information may also be extracted from multiple image sets. Image fusion allows aligning multiple datasets of the same modality over time or image sets from different modalities. The image sets can be simultaneously visualized and evaluated after image fusion. The segmentation process can also be based on information from multiple fused modalities.

### 2.3. Virtual Surgical Planning

Virtual surgical planning (VSP) is a simulation of the actual surgery on the imaging data [35]. It is based on information gathered in the previous treatment stages. The exact content of the virtual surgical planning depends on the type of implant. If a flat mesh plate is used, the virtual models of the mirrored orbit and affected orbit can be exported and 3D printed to serve as individual bending template(s) for molding the flat mesh. 

In the preformed implant setting, the stereolithographic model (STL) of a preformed implant is imported into the planning environment to perform a virtual reconstruction of the affected orbit. The implant’s fitting potential is evaluated and its optimal position for an accurate reconstruction of the pretraumatized anatomy is simulated. The potential of VSP in the preformed implant setting is thus highly dependent on the willingness of the implant manufacturers to provide STLs of their preformed implants. In modern planning software, the implant may be automatically aligned to another virtual model, for instance, the mirrored orbit. Manual adjustments could be necessary to prevent interferences with the bone and ensure the orbital defect is covered, with adequate implant support on the dorsal ledge and fixation possibility on the infraorbital rim. 

The implant may be virtually trimmed to simulate the cutting of medial or posterior parts of the implant. The surgery can be simulated multiple times in the virtual surgical planning, with different implant types and sizes (Figure 3). This enables comparison between preformed implant options and substantiated decision making before surgery. The number of try-ins in virtual planning is limitless without consequences for the patient, in contrast to try-ins during actual surgery. Establishing the optimal position in virtual planning provides the surgeon with intraoperative feedback, which could reduce the operating time and extent of manipulation inside the orbit during surgery [36].

### 2.4. Patient-Specific Implant Design

Reconstructing the orbit with a PSI is the ultimate step of individualization for orbital reconstruction. A PSI is virtually modelled from scratch, using information from the (advanced) diagnostics phase and exported virtual models. In dedicated design software, a prototype implant is generated. The prototype is imported into the virtual surgical planning and its fit is evaluated. The position of the prototype is not adjusted in the virtual surgical planning to improve the fit, but the design of the prototype is adjusted and the novel prototype is reimported. Even though the design of a PSI is not set in stone by protocols, various design considerations have been described in the literature. An overview of options is summarized in Table 1. This overview is not comprehensive, and novel design options are regularly introduced in the literature.

Design considerations can be categorized based on their intended effect: stability, positioning ease, accuracy of implant positioning or alleviation of clinical symptoms. The size and shape of the implant are dependent on the extent of the defect. The defect should be covered by the implant and its shape should reflect the intended reconstruction of the affected orbital walls. Support on existing bony structures is taken into account to ensure the stability of the reconstruction. Analogous to the preformed implants, support is most often found at three points in the orbit [22,38]. Fixation is recommended to ensure a stable position of the PSI [39,40]. Possible screw positions can be assessed in virtual planning, factoring in the patient’s anatomy and local bone quality. The thickness of the implant and implementation of an atraumatic cord around the edge are considerations that affect both the stability of the implant and the positioning ease during surgery. Due to additive-manufactured titanium’s rigidity, an implant thickness of 0.3 mm in combination with an atraumatic cord results in a good balance between rigidity and positioning ease.

The accuracy of implant positioning can be controlled by extensions over unaffected bony pillars. A compelling fit is created by the extension(s) of the implant over bony structures. An infraorbital rim extension limits rotation and translation in the anteroposterior direction [47]. Additional flanges to the posterior lateral wall may be implemented to prevent unwanted implant movement [43]. Screw positions from fixation material from a previous reconstruction can be re-used in secondary reconstruction to provide guidance and thus improve the accuracy of the implant positioning [54]. Another design option is to incorporate navigation markers and vectors, which can enhance the interpretation of feedback from the intraoperative navigation system. 

The last category, clinical symptoms, deals with the correction of globe displacement. The orbital volume is corrected to alleviate globe displacement, but the volume may be overcorrected to counteract fat atrophy and the anticipated iatrogenic loss of soft tissue [57]. The amount of overcorrection might be subjectively determined during surgery, by inserting additional spacers [51], or it may be fully integrated into the design of the PSI, posterior to the equator of the bulbus [22,38,50]. On the other hand, hypoglobus is the result of caudal displacement of the infra-orbital rim. An anterior elevation corresponding to the amount of downward displacement of the orbital rim may alleviate hypoglobus (Figure 4). The grid of the PSI can be designed with different techniques: using a large horizontal pattern to maximize drainage [37,40,43], or a more porous arrangement [42,45,46]. The multitude of design options and manual design leads to a wide range of possible PSI shapes (Figure 5).

The PSI design can be adapted to facilitate the reconstruction of multi-wall defects, for example, through the application of multiple PSIs (Figure 6). This enables a reconstruction that covers the entire defect while limiting the size of the PSI and, in turn, the required incision [46,55]. Depending on the connection used, it also provides the opportunity to create artificial support and relative feedback. A PSI that solely reconstructs the orbit will not suffice in cases with concomitant fractures of the surrounding bony structures. Repositioning the surrounding bone may be required in addition to the orbital reconstruction. Additional design options are available to gain feedback from PSIs on the subsequent reconstruction steps in these more extensive cases. An example of this is embedding the desired position of the zygomatic complex in the PSI design to facilitate correct repositioning [54].

### 2.5. Intraoperative Feedback

During surgery, the surgeon aims to position the implant as closely as possible to the ideal position that was established in the VSP. The availability of the VSP provides intraoperative feedback that improves the result of the reconstruction [35]. Additional types of feedback are available to aid in the accurate positioning of the implant (summarized in Table 2). The design options relating to implant positioning ensure static feedback through the unique and compelling fit of the PSI (Figure 7). In secondary cases, the re-use of screw positions from the primary reconstruction will also help to find the planned position.

Surgical navigation may be utilized to provide dynamic feedback on the implant position. During the registration for surgical navigation, the patient’s position in the operating room (OR) is linked to the preoperative imaging data in the virtual surgical planning. Several registration methods are available: soft-tissue registration, bone-anchored fiducials, and surgical splints [56,58]. Splint registration methods used to require repeated radiographic imaging with the fiducial splint in place, but the fusion of intraoral scan data in the advanced diagnostics phase allows the fabrication of a registration splint without additional radiologic imaging [59]. The splint is designed on the individual patient’s dentition and carries fiducials that can be indicated virtually in the planning software and physically in the OR. 

After registration, the navigation pointer’s position in the patient is visualized in the virtual surgical planning on the navigation system’s screen. This provides the surgeon with feedback on the position of the pointer, representing the position of the indicated location (a specific spot on the implant’s surface). The quality and interpretability of the feedback may be enhanced through navigation markers embedded in the design [39,52]. The markers are indicated in the VSP as navigation landmarks and used as a reference in the OR. If the surgeon positions the pointer in the navigation marker on the implant, visual and quantitative feedback about the pointer’s position compared to the landmark is provided.

### 2.6. Evaluation

Intraoperative imaging of the patient after implant positioning is highly recommended, since the realized implant position can be qualitatively evaluated. If the surgeon is not satisfied with the position, the implant can be repositioned in the same surgical setting, preventing a revision surgery. For a complete quantitative evaluation of the surgical result, the scan can be reconstructed and fused with the VSP. The planned position of the implant can be compared to the achieved position of the implant to enable an objective assessment (Figure 8). Differences between planned and realized position can be quantified in three dimensions and expressed as rotations (roll, pitch and yaw) and translations (x, y and z) [60]. Additionally, the volumetric difference between the reconstructed and unaffected or planned orbit may be assessed. Post-operative CT-scans are indicated if intra-operative CT is not acquired (or offered incomplete information), or if clinical considerations in the follow-up period necessitate additional imaging.

An optimally positioned orbital implant is no guarantee for a perfect clinical outcome. Restoration of the globe position can be relatively well achieved with a PSI, even in secondary reconstructions [22]. It is more complex to treat diplopia, as it involves the mechanical mobility of the eye, combined visual perception, and processing in the visual cortex. Visual processing may (partially) adapt over time. At discharge, the patient is informed that double vision will be experienced for the first 10–14 days, possibly longer. Ocular motility can be improved by training the extraocular muscles to prevent scarring and anticipate fibrosis [61]. Instructions are provided to mobilize the eye as much as possible: monocular orthoptic exercises six times per day for 6–12 weeks to prevent adhesions and stimulate a reduction in orbital soft tissue swelling, especially for the extraocular muscles. This protocol positively affects clinical improvement in both primary and secondary cases [13,22,47].

## 3. Discussion

Surgical complexity and the fact that an inappropriate reconstruction potentially leads to an adverse outcome have led to the parallel incorporation of computer-assisted surgery and personalized medicine in the orbital reconstruction workflow. Although both concepts aim to optimize treatment outcome, their rationale differs. CAS centralizes medical technology to improve the predictability and accuracy of all treatment stages, but various steps have been standardized and can be considered independent of the patient. CAS has catalyzed the incorporation of personalized medicine in orbital reconstruction: virtual surgical planning technology enables surgical preparation, implant selection, and the simulation of the desired implant position based on the individual’s characteristics. The concepts are considerably intertwined in PSI reconstruction, and mutual interactions can be discerned. Personalization of the implant design is directly affected by information gathered in the preoperative CAS stage, and tailoring the implant to the patient’s anatomy provides feedback on positioning during the intraoperative stage. Intraoperative CAS technology supports the accurate positioning of the PSI, which is a prerequisite for achieving the intended treatment effect of the personalized implant. In light of this symbiosis, surgical reconstruction with a PSI can be considered the pinnacle of both CAS and personalized medicine in orbital reconstruction.

Several larger comparative studies have demonstrated the beneficial effects of (components of) CAS on the accuracy of volumetric reconstruction [62], clinical outcome [36], and need for revision surgery [63]. In practice, a combination of several CAS building stones is often utilized. This yields heterogeneity in surgical approaches, which makes it difficult to compare outcomes between studies. Differences in indication, patient and fracture characteristics, or implant materials used further complicate comparison [64]. Isolating the effect of individual CAS techniques on patient outcome is hampered by an overlap of techniques within study populations. The individual effects of CAS techniques have been assessed in a one-to-one comparison in a cadaver series [65]. Despite limitations associated with the cadaver model and an inability to assess clinical outcome parameters, positive effects of virtual planning, intraoperative imaging, and surgical navigation on reconstruction accuracy were established.

Several indications for PSI in orbital reconstruction have been advocated, relating to defect size, location, or timing of reconstruction [22,47,49,50,66,67,68,69]. The common denominator between extensive defect size, the lack of bone support in the posterior third of the orbit, or secondary reconstruction after inadequate primary reconstruction is that the difficulty of reconstruction has been significantly increased. The possibility to perfectly tailor the shape of the implant to the patient’s anatomy makes a PSI better suited for these complex reconstructions compared to implants that lack these options. Other advantages of PSIs are improved ease of use and precise, accurate fit, which leads to accurate implant positioning and a decreased surgery time [41,43,45,49,50,70,71,72]. Iatrogenic soft-tissue damage is prevented as much as possible since the number of try-ins necessary to correctly position the implant is reduced, and no sharp edges are present around the implant’s circumference [37,72,73,74]. These factors lead to the high predictability of functional and aesthetic outcomes, fewer complications during or after surgery, and a lower revision rate than other implant types [37,45,54,62,67].

The list provided in Table 1 stipulates an ever-increasing armamentarium of design features to improve (ease of) positioning or alleviate clinical symptoms. The positioning-related characteristics guarantee adequate implant support and a unique, compelling fit for the implant. It is vital to evaluate the surgical accuracy of the PSI reconstruction and the added value of the positioning features [72]. Several studies have used a comparison between unaffected and reconstructed orbital volume or angulation between the floor and medial to measure surgical accuracy [37,75,76,77]. These outcome parameters may conceal incorrect implant positioning. In contrast, a direct comparison between planned and acquired implant positions will reveal all surgical errors qualitatively and quantitatively [60]. Assessing individual degrees of freedom is even feasible with this approach. The evaluation phase is an indispensable component of the CAS workflow and should be performed for each orbital reconstruction. A direct comparison between planned and acquired implant positions is advocated to maximize the potential of the evaluation phase. 

Incorporating clinical considerations in the design is currently not as clear as the positioning-related features. Since the early introduction of PSI reconstruction, the overcorrection of resulting orbital volume has been suggested several times, and its use has been described (see Table 1). Still, guidelines to the amount of overcorrection are arbitrary and subjective, and not substantiated by evidence. Information about the location and shape of the overcorrection is lacking. The overcorrection may even be introduced through different pathways: it may be embedded in the PSI design or added afterwards through titanium blocks [51]. While the second option provides some freedom to the surgeon intraoperatively, the judgment is subjective and hampered by the soft-tissue reaction to the surgery (and the trauma in a primary setting). Trial and error positioning leads to the increased manipulation of orbital tissue, and any dislocation of the blocks may warrant a second procedure [78,79]. Embedding overcorrection in the design is suggested to be the best solution for achieving an optimal result [78] and could be precisely tailored to the individual patient in the future, provided the abovementioned knowledge gaps are filled. 

Cost, lead time, and logistic demands are drawbacks of using a PSI [37,71,73,79]. Pricing may vary based on geography, but the process usually costs EUR 1500-6000 [57,72,80,81]. Manufacturing of the implant takes approximately 3–5 working days; this does not include sterilization, or the time needed for virtual surgical planning and design. Korn et al. described a mean communication time between the surgeon and the PSI company technician during virtual surgical planning of almost nine days for isolated wall fractures and 16 days for multi-wall fractures [82]. An adjustment to the initial design that the technician proposed was necessary in nearly three-quarters of cases, but implants from technicians with previous in-house training required fewer adjustments. Improved communication and mutual understanding are suggested to be the reasons for the efficiency improvement. Complete in-house planning and design by a dedicated, on-site technician may ameliorate planning efficiency, ultimately greatly reducing the lead time (provided the surgeon and technician are experienced and have cooperated on previous cases). In-house design is suggested to reduce costs, since commercial partners are only relied upon for fabrication [81]. These benefits of in-house design may be why surgeons who use in-house planning feel less hindered by the drawbacks of using a PSI [71].

Although this paper focuses on post-traumatic orbital reconstruction, other PSI applications relating to the orbit have also been described. In zygomatic reconstruction after trauma, ablative surgery, or congenital malformation, PSIs were found to precisely restore the anatomy without the need for additional bone grafts [83]. In secondary post-traumatic reconstruction of the orbit and zygoma, PSIs enable a one-stage surgical procedure in which the surgical order is reversed: by operating the orbit first, the functional result of the orbital reconstruction is independent of repositioning the zygomatic complex [54]. PSIs may also be used to create an artificial orbital rim and floor for globe support after maxillectomy [84,85]. The most extensive orbital PSI reconstructions have been described after the resection of a spheno-orbital meningioma or neurofibroma [55,86]. In these cases, the reconstruction of all four orbital walls with multiple PSIs enabled a predictable reconstruction of the internal orbit in the same surgical setting as the resection. The PSI design in the abovementioned cases could differ greatly from the design in the post-traumatic reconstruction of solitary orbital fractures. Still, the rationale behind using a PSI is the same: freedom of design to adapt the PSI to the patient’s anatomy and a predictable and accurate final result.

## 4. Conclusions

An overview of the CAS workflow for post-traumatic orbital reconstruction has been presented, with an in-depth description of the techniques embedded in the workflow and a special focus on PSI. It has been demonstrated how the conventional workflow can be complemented by both CAS and personalized medicine in order to optimize the clinical outcome of post-traumatic orbital reconstruction. CAS technology has catalyzed the incorporation of personalized medicine in orbital reconstruction. The reconstruction of the orbit with a PSI can be considered the pinnacle of CAS and personalized medicine. There are no strict guidelines for the design of patient-specific implants, but several design considerations can be implemented to improve the positioning or alleviation of clinical symptoms. Because of the high predictability of aesthetic and functional outcomes, the use of PSIs has been advocated especially in difficult reconstructions. Cost, lead time, and logistical demands are known drawbacks, although they may be alleviated by in-house design.

## Figures and Tables

**Figure 1 jpm-12-01366-f001:**
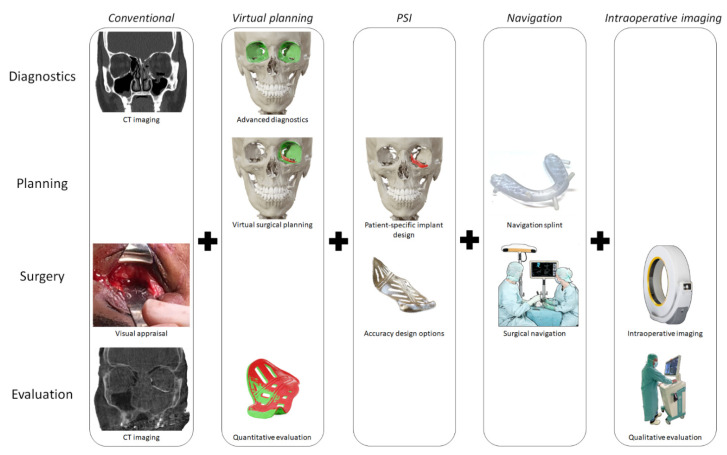
Schematic overview of the technological possibilities within the orbital reconstruction workflow. Chronologically, a clinical workflow consists of diagnostics, planning, surgery, and evaluation. In the conventional workflow, the surgeon is dependent on clinical assessment and preoperative imaging, visual appraisal during surgery and postoperative imaging for evaluation. CAS consists of several techniques that may be combined and affect one or multiple workflow stages. Several CAS technologies may be combined.

**Figure 2 jpm-12-01366-f002:**
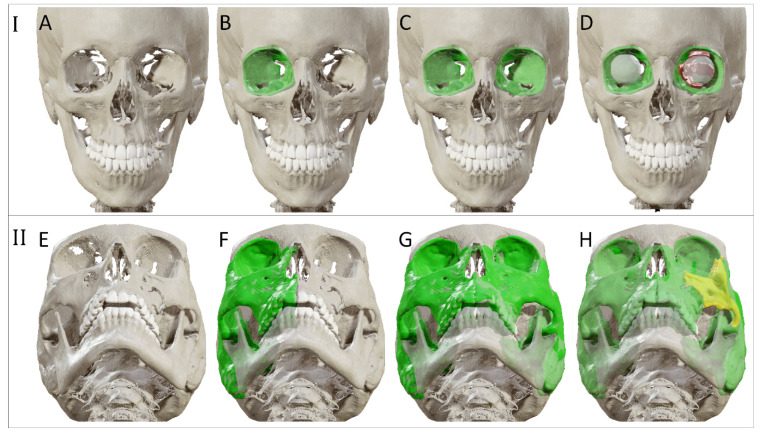
Advanced diagnostics for two cases. I: Solitary orbital reconstruction (**A**–**D**). (**A**) Visualization of the 3D bone surface model. (**B**) Segmentation of the unaffected orbit. (**C**) Mirroring of the segmented orbit to the affected, contralateral side. (**D**) Visualization of additional structures, such as the globe and eye muscles. II: Zygomatic complex fracture (**E**–**H**). (**E**) Visualization of the 3D bone surface model. (**F**) Segmentation of the unaffected side. (**G**) Mirroring of the segmentation to the affected, contralateral side. (**H**) Visualization of the zygomatic complex displacement.

**Figure 3 jpm-12-01366-f003:**
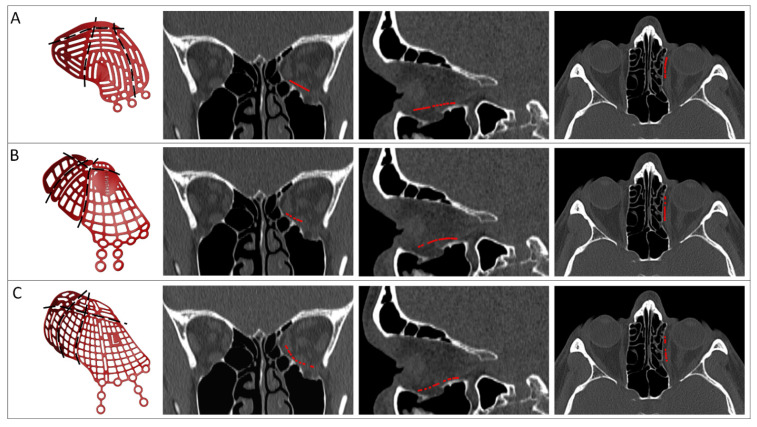
Virtual fitting of different preformed implants in a solitary orbital fracture. Three-dimensional models of the preformed implants of KLS Martin (**A**), Synthes (**B**) and Stryker (**C**) are visualized with potential cutting lines (black lines) in the first column. The implants are virtually positioned (red contour) and the fit is evaluated in the coronal, sagittal, and axial slices. Important considerations are adequate support (on the posterior ledge, on the medial wall and on the infraorbital rim) and a lack of interference with bone.

**Figure 4 jpm-12-01366-f004:**
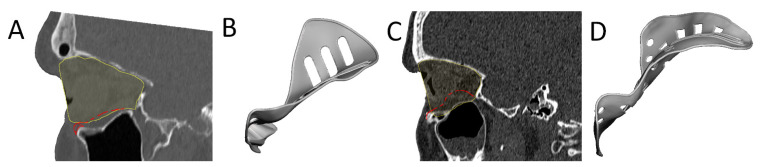
Examples of two patient-specific implant designs with overcorrection (red contour) of the mirrored orbital volume (yellow contour). The first patient-specific implant is designed with an anterior elevation at the infra-orbital rim to compensate for the asymmetry in globe position (**A**,**B**). The second patient-specific implant is designed with a large overcorrection for an anophthalmic socket reconstruction (**C**,**D**).

**Figure 5 jpm-12-01366-f005:**
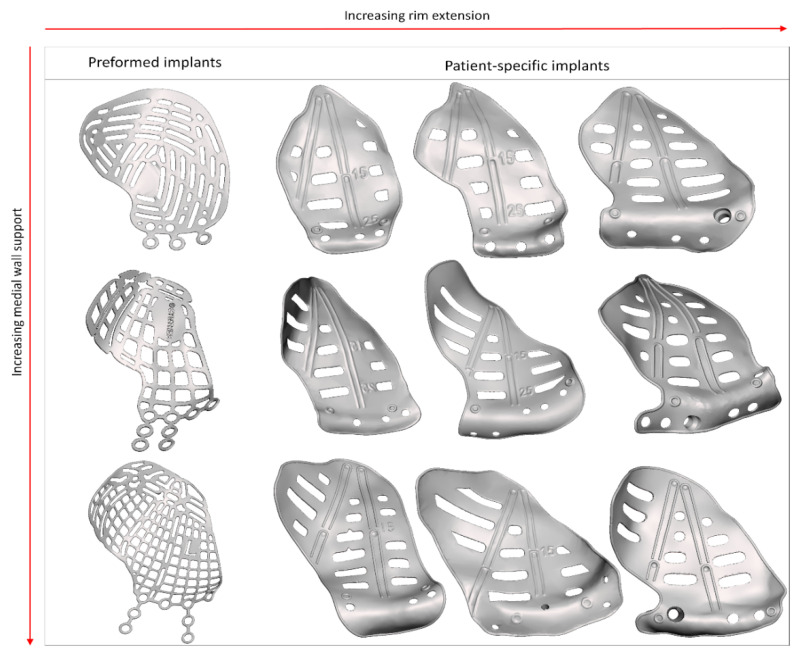
Different shapes of the available preformed implants and patient-specific implants are illustrated. There is a wide variety in shapes in the patient-specific implants. From left to right the rim extension is increased. From top to bottom the medial wall support is increased.

**Figure 6 jpm-12-01366-f006:**
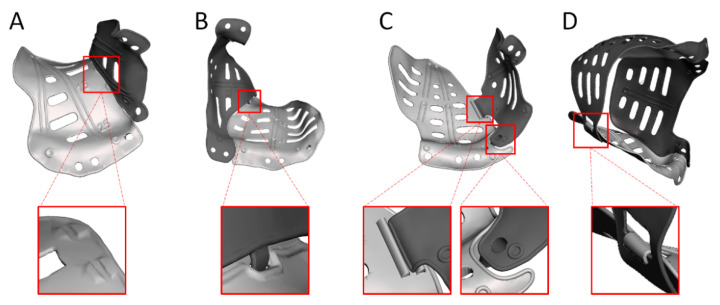
Patient-specific implant design for multi-wall cases. (**A**) Ridges on the orbital floor implant provide relative feedback for the positioning of the lateral wall implant. (**B**) Patrix–matrix connection to connect a medial wall and orbital floor implant. (**C**) The orbital floor implant with medial wall extension is connected to a lateral wall implant dorsally with ridges and anteriorly with a puzzle connection. (**D**) Four-wall reconstruction with a hook-and-bar connection for additional support for the orbital floor implant.

**Figure 7 jpm-12-01366-f007:**
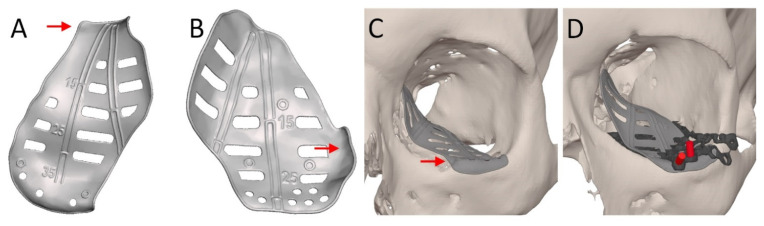
Illustration of the different feedback methods.). Markers and vectors are visualized in (**A**) and (**B**). Compelling fit of the patient-specific implant design is indicated with red arrows (**A**–**C**) Segmentation of screw holes of the previous reconstruction are illustrated in red (**D**) and the previous implant in dark grey.

**Figure 8 jpm-12-01366-f008:**
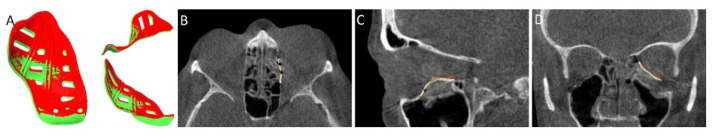
Illustration of an evaluation. (**A**) Three-dimensional model of the planned patient-specific implant (red) and the realized patient-specific implant (green) viewed from different perspectives. (**B**–**D**) Axial, sagittal, and coronal view of the postoperative computed tomography scan with the planned contour of the patient-specific implant in red.

**Table 1 jpm-12-01366-t001:** List of different design considerations as reported in the literature.

Design Consideration	Effect on	Options	References	Notes
Thickness	Positioning, stability	0.3 mm	[22,37,38]	
Atraumatic cord	Positioning, stability	Present	[37,39,40]	
		Absent	[38,41,42]	
Grid	Clinical symptoms	Horizontal	[22,37,40,43]	
		Squares	[38,39,41,44]	
		Porous	[42,45,46]	
Support	Stability, accuracy	Three points	[22]	Infraorbital rim, medial wall, posterior ledge
			[38]	Anteromedial, anterolateral, posterior
		Ledge	[37,40,43]	Inverted shovel design
		Lateral posterior wall	[43]	Stabilizer for self-centering implant
Extension	Accuracy	Orbital rim	[22,42,44,46,47,48]	
		Lateral posterior wall	[43]	
		Specific bone features	[45]	
Anterior elevation	Clinical symptoms		[22]	Rim elevation to correct hypoglobus
Overcorrection	Clinical symptoms	Location	[22]	Posterior to bulbus
			[49]	Orbital floor elevated in sagittal relation
		Amount	[22]	Based on clinical findings, advanced diagnostics
			[38]	Slight overcorrection
			[50]	Same amount in cubic cm as mm enophthalmos
		Intraoperatively	[51]	Spacers
Navigation	Accuracy	Markers	[22,37,38,39,52]	Eminence lacrimal foramen [38]
		Vectors	[37,40,43]	
Fixation	Stability	Absent	[38,44,48]	
		Present	[22,37,39,40,42,46,47,53]	Eccentric screw alters implant position [47] Fix implant if form stable [40]
Fixation re-use	Accuracy	Re–used screw hole	[54]	Only in secondary reconstruction
Multi-piece	Positioning, stability,	Lazy-S	[42,47,49]	
	accuracy	Interlocking	[46,48,55,56]	

**Table 2 jpm-12-01366-t002:** Different feedback methods available in the operation room.

Feedback Method	Static/Dynamic
Virtual surgical planning	Static
Compelling fit patient-specific implant	Static
Fixation re-use	Static
Navigation	Dynamic
Markers and vectors	Dynamic
Intraoperative imaging	Static

## Data Availability

Not applicable.

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
