# Peer review of "Personalized Medicine Workflow in Post-Traumatic Orbital Reconstruction"

_jpm, 2022, doi:10.3390/jpm12091366_

Round 1

Reviewer 1 Report

Thank you for the opportunity to review the article "Current status of personalized medicine in post-traumatic orbital reconstruction". 

The topic is relevant and up to date. The article addresses an important topic for OMF surgery. However, it is difficult for the reader who is not well informed in the field to derive any added value from this article, as some theses are postulated quite out of context. For example, the impression is that only the PSI workflow is state of the art, and the digital workflow is described without significantly elaborating on the differences to conventional procedures. Flowcharts or visual comparisons of the supply variants would certainly be helpful here. To increase the quality of the article, significant changes should be made:

- A linguistic, and grammatical revision by a native speaker is recommended. Many formulations do not correspond to the standard.

- The abstract is incomprehensible and should be summarized. It is not apparent to the reader at first glance what the goal of this review paper is. A clear outline is needed!

- Indications of care should be better presented.

- Line 37: Pathogenesis of an orbital fracture is too generally explained as "energy is transferred ...". There are different approaches that explain a fracture.

- Likewise, with the explanation of soft tissue injury. This should be more clearly stated.

- The goal of reconstruction is certainly not just "restoring bony contours".

- Is symmetry the goal of orbital reconstruction? How many people are symmetrical?

- ...

- Classical structure for literature review not available (see PRISMA Guidelines).

... Apart from that, an exciting article which, after some revision, will undoubtedly add value to this field.

Author Response

Dear Editor,

We have received the reviewers’ comments on the manuscript: “Current status of personalized medicine in post-traumatic orbital reconstruction”. We would like to thank the reviewers for the time they committed to reviewing the manuscript and their valuable comments. The response to the reviewers’ comments is found below.

We would like to take this opportunity to clarify the intentions behind the submission. The submission has been the result of a manuscript invitation for a special issue about utilization of 3D personalized surgery, with the goal of demonstrating the current state of the art techniques (with a peak into the future). unfortunately, the option “invited manuscript” was not provided in the article type list; review seemed to be the closest available option. This has led to some (justified) comments about the methodology of literature search and manuscript selection for a literature review. We have adjusted the article type and altered the manuscript title for clarification purposes.

Yours faithfully,

J.F. Sabelis, MSc

  1. Schreurs, MSc, PhD
  2. Essig, MD, DDS, PhD

A.G. Becking, MD, DDS, PhD, FEBOMS

  1. Dubois, MD, DDS, PhD

Reviewer 1

English language and style

( ) Extensive editing of English language and style required

(x) Moderate English changes required

( ) English language and style are fine/minor spell check required

( ) I don't feel qualified to judge about the English language and style

Is the work a significant contribution to the field?                                                                           3/5        

Is the work well organized and comprehensively described?                                                     3/5

Is the work scientifically sound and not misleading?                                                                      3/5

Are there appropriate and adequate references to related and previous work?                 3/5

Is the English used correct and readable?                                                                                           3/5

Comments and Suggestions for Authors

Thank you for the opportunity to review the article "Current status of personalized medicine in post-traumatic orbital reconstruction".

The topic is relevant and up to date. The article addresses an important topic for OMF surgery. However, it is difficult for the reader who is not well informed in the field to derive any added value from this article, as some theses are postulated quite out of context. For example, the impression is that only the PSI workflow is state of the art, and the digital workflow is described without significantly elaborating on the differences to conventional procedures. Flowcharts or visual comparisons of the supply variants would certainly be helpful here.

We agree with the reviewer that the topic is important for OMF surgery. The aim of the submission was to demonstrate benefits of the 3D personalized workflow within orbital reconstruction to a wider audience, but useful suggestions are provided by the reviewer to improve this. PSI is considered the pinnacle of computer-assisted surgery and personalization in orbital reconstruction, but certainly not the only state-of-the-art option; several computer-assisted state-of-the-art options are discussed in the manuscript. Figure 1 has been replaced by an overview of the different variants of computer-assisted surgery in orbital reconstruction, based on the reviewer’s suggestion. Differences between conventional and computer-assisted workflows have been highlighted in the corresponding description.

To increase the quality of the article, significant changes should be made:

- A linguistic, and grammatical revision by a native speaker is recommended. Many formulations do not correspond to the standard.

Linguistic and grammatical revision of the original manuscript has been performed.

- The abstract is incomprehensible and should be summarized. It is not apparent to the reader at first glance what the goal of this review paper is. A clear outline is needed!

The abstract has been rewritten.

- Indications of care should be better presented.

Surgical indication is a controversial topic within orbital fracture management. Some surgeons advocate a conservative approach with delayed surgery if clinical symptoms develop, while others choose a more aggressive approach, anticipating on development of clinical symptoms. Systematic reviews show large heterogeneity in literature and limited scientific evidence on relative indications. A detailed overview of the discussion surrounding the topic is outside the scope of this submission about the 3D personalized workflow for orbital reconstruction. Nevertheless, some information on indication has been added to the introduction (“There is an.. and patient characteristics.” Line 56-61). In addition, throughout the text, references to indication for surgical repair were already described (“Absolute restrictions, as … through repeated measurements.”, “Shape alterations of … for surgical reconstruction.”).

- Line 37: Pathogenesis of an orbital fracture is too generally explained as "energy is transferred ...". There are different approaches that explain a fracture.

The authors agree that this is a very minimalistic description of the pathogenesis. A description of the buckling theory and hydraulic theory are added to the introduction.

- Likewise, with the explanation of soft tissue injury. This should be more clearly stated.

The pathogenesis of soft-tissue effects relevant to an orbital fracture are provided in the third paragraph of the introduction (“The orbital volume…in clinical presentation”).

- The goal of reconstruction is certainly not just "restoring bony contours".

This is correct, and “restoring bony contours” is not the complete description that was provided (“restoring the bony contours to support the soft tissue and repositioning the globe to recover ocular function”). An additional comment is provided to explain the importance of restoring the bony contours (“Restoration of the orbit is the first and most predictable step in surgical management of orbital fractures”).

- Is symmetry the goal of orbital reconstruction? How many people are symmetrical?

Symmetry in itself is not the goal of orbital reconstruction, but the unaffected orbit is the best reference for the affected orbit (as described in the advanced diagnostics section). This approach is substantiated by scientific evidence on the high level of orbital symmetry. A remark on orbital symmetry has been added to the advanced diagnostics section (“The volume of the affected orbit can be compared to the unaffected healthy side to determine the relative volume change, since it has been proven that the orbits are highly symmetrical .”).

- Classical structure for literature review not available (see PRISMA Guidelines).

The reviewer is correct that the PRISM structure is not available, nor any alternative approach is described. The manuscript was submitted after an invitation for a special issue on 3D personalized surgery. The goal was to demonstrate the current state-of-the-art approach in orbital reconstruction, rather than provide a systematic review on one of the subtopics discussed (patient-specific implants, virtual surgical planning, computer-assisted surgery, personalized medicine in orbital reconstruction). The outline and intention of the manuscript are clarified in the introduction section of the revised manuscript to prevent future confusion.

... Apart from that, an exciting article which, after some revision, will undoubtedly add value to this field.

We thank the reviewer for their positive overall closing remark and the remarks brought forward that made significant improvement to the manuscript possible.

Reviewer 2 Report

The authors have submitted a review of the modern approaches to post-traumatic orbital reconstruction. In their paper they describe they discuss the workflow of orbital reconstructions, emphasizing the three stages of computer-assisted surgery and how this relates to personalization of the reconstruction itself: virtual surgical planning, intraoperative guidance, and postoperative evaluation. They also discuss the broad categories of orbital implant options: flat implants, preformed implants, and patient specific implants.

The main strength of the manuscript is the breadth of discussion on the technicalities of the surgical planning, execution, and evaluation for orbital reconstruction. The authors have provided a lot of information that other surgeons may implement into their own practices. They also provide good-quality images to elaborate on some of the contents of the manuscript, albeit with some exceptions noted below. The scope of this review is also very broad and informative, and the amount of references collected indicate that the authors have reviewed a large amount of literature in preparing this review.

The main weakness of the manuscript is the general readability of the manuscript. The organization of sub-headings, formatting, and word flow make this manuscript difficult to follow. Some of the Figures do not add valuable information to the paper, and Table 1 offers little information relative to the amount of space it takes. The manuscript is quite lengthy and could easily be shortened with the removal of some redundant components. Namely, the Introduction is long and includes too much generalized information that is not relevant to the intended purpose of the manuscript. The Abstract does not give the reader a sense of what type of study is being presented as well as what kind of information to anticipate in the manuscript. The Methods section is absent and even though this paper is labeled as a Review, it should still include some information regarding how the information was collected, assessed, interpreted, and included in the final manuscript. Lastly, and perhaps most importantly, there is no discussion of outcomes with the approaches described in this paper. The authors include some comments on what they anticipate to be the benefits with some approaches over others, but this is not sufficient when there is likely more outcome data available in the literature they have reviewed. Outcome information is highly relevant in the context of deciding whether to pursue certain approaches, especially when some are notably more expensive and complicated.

There has been a lot of growth in orbital reconstruction knowledge over the past decade, especially in regard to personalization of the approach for individual patients. Thus, this topic would be a great addition to the audience of the Journal of Personalized Medicine. However, the manuscript, requires additional work and should be reconsidered after major revision.

Specific comments/edits/suggestions/questions are below:

Introduction

·         Line 28: The plural form of “trauma” would be more appropriate.

·         Line 30-36: The authors should consider elaborating on the connective tissues of the orbit, rather than its soft tissues. All that is mentioned about the connective tissues is the “hard shell” comment, which is deserves more elaboration on the actual anatomy of the orbital connective tissue.

·         Line 39-41: This statement is not correct. Enophthalmos and hypoglobus are only two of several possible forms of displacement following trauma. Virtually any form of displacement is possible.

·         Line 43: The sentence would be more grammatically correct without the word “causing”.

·         Line 44: “Connective tissue” is a more appropriate term than “hard tissue”.

·         Line 64: The word overview could be replaced with the term “visualization”.

·         Line 71: “Virtual surgical planning (VSP), intraoperative guidance, and postoperative evaluation.” — This list should follow a colon from the preceding sentence.

·         The authors should be clear about their purpose and intent of the paper.

·         The Introduction is long and contains a lot of generalized information that is readily available elsewhere. The manuscript would be easier to read and the focus of the paper (i.e. surgical approaches) would come across better if the Introduction was shortened.

Figure 1

·         This Figure does not add much to the paper and the authors should consider removing it to be more concise.

2.1 Diagnostics

·         Line 102: “enophthalmos, hypoglobus” should be replaced with a more general term, such as “globe displacement”, as these are not the only forms of displacement that are possible.

·         Line 104: It should be specified that the Hertel mainly only measures relative *anterior-posterior* position. It does not necessarily measure displacement in other axes.

·         Line 107: “There are no readily available reproducible measurement tools to measure hypoglobus” — This might be correct in terms of *clinical tools*, but the CT scan is able to accurately and reproducibly measure globe displacement.

·         Line 108: The term for this technique is the “Hirschberg Test”.

·         Line 115: “Impingement” is one of several possibilities. The EOMs can be severed and may also be entrapped by fractured bone.

·         Line 115: Any of the EOMs can be impinged and cause diplopia, so it is not advisable to specify this for only the “inferior ocular muscles”.

·         Line 119: It would be helpful to specify how soon the operations should take place in these cases.

2.3 Virtual Surgical Planning

·         Line 169: It would be beneficial to list some indications for surgical reconstruction.

2.3.1 Patient-specific implant design

·         Line 201-203: “The prototype is imported into the virtual surgical planning and its fit evaluated; in contrast to the preformed implant setting, the position of the prototype is not adjusted, but its design.”— This sentence should be reformatted into 2 sentences (semi-colon should be replaced with a period). The second half of the sentence, after the semi-colon, does not read well and should be reworded.

·         Line 235: It would be beneficial to elaborate on how the orbital volume is corrected. This is not always achieved by the implant alone.

·         Line 236: “anticipated” is more correct.

Table 1

·         This Table is a great idea to include, but it should be modified from its current form. The Table title should probably use the term “design considerations” instead of “design options”. The table should provide additional information in the Notes column (i.e. more than just the quotations from the referenced papers). There is no need for a reference column since it takes up too much space and the references can still be added elsewhere in the table without the full name of the author.

Figure 6

·         This is a useful Figure. However, if there are labels (i.e. “A,B,C,D”), they should be described in the Figure legend.

·         The final sentence, “From top to bottom” does not make sense with the given image.

2.4 Intraoperative Feedback

·         Line 294: “e.g. a specific spot on the impolant” should be in parentheses.

2.5 Evaluation

·         Line 315: Is it correct to say that post-operative CT scans are *only* indicated when intra-operative CT scans were not performed or offered incomplete information?

·         Line 328-329: “Interestingly, ocular motility can be improved by training the extraocular muscles to prevent and anticipate fibrosis.” — This is controversial as it is not widely established. The authors should include elaboration and references.

Author Response

Dear Editor,

We have received the reviewers’ comments on the manuscript: “Current status of personalized medicine in post-traumatic orbital reconstruction”. We would like to thank the reviewers for the time they committed to reviewing the manuscript and their valuable comments. The response to the reviewers’ comments is found below.

We would like to take this opportunity to clarify the intentions behind the submission. The submission has been the result of a manuscript invitation for a special issue about utilization of 3D personalized surgery, with the goal of demonstrating the current state of the art techniques (with a peak into the future). unfortunately, the option “invited manuscript” was not provided in the article type list; review seemed to be the closest available option. This has led to some (justified) comments about the methodology of literature search and manuscript selection for a literature review. We have adjusted the article type and altered the manuscript title for clarification purposes.

Yours faithfully,

J.F. Sabelis, MSc

  1. Schreurs, MSc, PhD
  2. Essig, MD, DDS, PhD

A.G. Becking, MD, DDS, PhD, FEBOMS

  1. Dubois, MD, DDS, PhD

Reviewer 2

English language and style

( ) Extensive editing of English language and style required

(x) Moderate English changes required

( ) English language and style are fine/minor spell check required

( ) I don't feel qualified to judge about the English language and style

Is the work a significant contribution to the field?                                                                           3/5

Is the work well organized and comprehensively described?                                                     2/5

Is the work scientifically sound and not misleading?                                                                      3/5

Are there appropriate and adequate references to related and previous work?                 4/5

Is the English used correct and readable?                                                                                           4/5

Comments and Suggestions for Authors

The authors have submitted a review of the modern approaches to post-traumatic orbital reconstruction. In their paper they describe they discuss the workflow of orbital reconstructions, emphasizing the three stages of computer-assisted surgery and how this relates to personalization of the reconstruction itself: virtual surgical planning, intraoperative guidance, and postoperative evaluation. They also discuss the broad categories of orbital implant options: flat implants, preformed implants, and patient specific implants.

The main strength of the manuscript is the breadth of discussion on the technicalities of the surgical planning, execution, and evaluation for orbital reconstruction. The authors have provided a lot of information that other surgeons may implement into their own practices. They also provide good-quality images to elaborate on some of the contents of the manuscript, albeit with some exceptions noted below. The scope of this review is also very broad and informative, and the amount of references collected indicate that the authors have reviewed a large amount of literature in preparing this review.

We thank the reviewer for their positive comments on the manuscript. Providing information to surgeons on the complete extent of topics associated with the 3D personalized workflow in orbital reconstruction and providing an in-depth description of the workflow to ease implementation were among the main goals behind the overview presented in the manuscript. A large amount of literature was used to provide a solid scientific base for topics described.

The main weakness of the manuscript is the general readability of the manuscript. The organization of sub-headings, formatting, and word flow make this manuscript difficult to follow. Some of the Figures do not add valuable information to the paper, and Table 1 offers little information relative to the amount of space it takes.

We agree with the reviewer’s comment on Table 1: a new formatting has considerably shortened the table’s length and improved its readability. Figure 4 has been omitted and Figure 1 has been replaced by an overview of the conventional and potential computer-assisted workflows. The levels of sub-headings were reduced, with only two levels remaining.

The manuscript is quite lengthy and could easily be shortened with the removal of some redundant components. Namely, the Introduction is long and includes too much generalized information that is not relevant to the intended purpose of the manuscript. The Abstract does not give the reader a sense of what type of study is being presented as well as what kind of information to anticipate in the manuscript.

The introduction has been considerably shortened. The aspects remaining in the introduction might be generalized information for readers familiar with orbital reconstruction, but are deemed necessary information for the wider audience interested in 3D personalized medicine. The abstract has been rewritten to provide a better summary of the manuscript’s goals and content.

The Methods section is absent and even though this paper is labeled as a Review, it should still include some information regarding how the information was collected, assessed, interpreted, and included in the final manuscript.

The paper was indeed labeled as a review upon submission. As explained in the general comments to the Editor, invited manuscript was unfortunately not available in the article type selection. As explained above, the aim was to provide an overview of (all aspects of) the 3D personalized workflow in orbital reconstruction. Although this overview was supported by literature, the intention of the manuscript was not to provide a systematic review on the literature available on one specific subtopic of computer-assisted surgery. The goal of the manuscript has been clarified in the abstract and introduction and the manuscript has been labeled ‘invited manuscript’ to prevent confusion.

Lastly, and perhaps most importantly, there is no discussion of outcomes with the approaches described in this paper. The authors include some comments on what they anticipate to be the benefits with some approaches over others, but this is not sufficient when there is likely more outcome data available in the literature they have reviewed. Outcome information is highly relevant in the context of deciding whether to pursue certain approaches, especially when some are notably more expensive and complicated.

Outcome evaluation is indeed very important to establish the possible benefits and weigh them against the known drawbacks of increased cost and complexity. Since CAS consists of different building stones (as shown in Figure 1), there is a large variety on workflows that are utilized in literature. Often, the use of technologies in study populations overlap. A case in point is the largest comparative study published to date (Zimmerer et al.), where individualized implants are compared preformed implants, but there is a significant difference in the use of navigation (and quite possibly the use of virtual surgical planning) between the groups. This hampers assessment of the effect of individual CAS techniques on clinical outcome. Heterogeneity in indication, timing, fracture complexity, implant types and selection bias further hamper a definitive answer on the benefits of individual technologies. A large cadaver study on reconstruction accuracy has been undertaken by the authors, which established a positive effect of virtual planning, intraoperative imaging and surgical navigation.

A paragraph has been added to the discussion section about the beneficial effect of CAS that have been described in literature, with remarks about the known limitation of establishing the individual technique responsible and the heterogeneity in approaches and populations (“Several larger comparative… accuracy were established”). The results of the cadaver study and the known limitations of this approach are also provided in this paragraph.

The beneficial effects of PSI reconstruction have been more frequently described in literature. A summary of the outcome information was provided in the discussion as well (“Several indications for…other implant types”).

There has been a lot of growth in orbital reconstruction knowledge over the past decade, especially in regard to personalization of the approach for individual patients. Thus, this topic would be a great addition to the audience of the Journal of Personalized Medicine. However, the manuscript, requires additional work and should be reconsidered after major revision.

We agree that the topic is a great fit for the audience of Personalized Medicine and that the revisions based on the reviewers’ comments fulfill the requirements towards a publishable manuscript.

Specific comments/edits/suggestions/questions are below:

Introduction

Line 28: The plural form of “trauma” would be more appropriate.

In the revised manuscript, the complete sentence has been revised.

Line 30-36: The authors should consider elaborating on the connective tissues of the orbit, rather than its soft tissues. All that is mentioned about the connective tissues is the “hard shell” comment, which is deserves more elaboration on the actual anatomy of the orbital connective tissue.

We agree that an elaboration should be provided, especially for readers less familiar with orbital reconstructive surgery. This has been adjusted in the revised manuscript (“The orbit is an inward-projecting bony structure in the shape of a cone (or pyramid) at the transition between midface and skull base. The base of the orbit, the orbital rim, is composed of thick bone; in contrast, the orbit’s inner walls are thin bony structures. The orbit provides the casing for the soft-tissue structures associated with the visual (motor) system”)

Line 39-41: This statement is not correct. Enophthalmos and hypoglobus are only two of several possible forms of displacement following trauma. Virtually any form of displacement is possible.

Any form of displacement is indeed possible, but enophthalmos and hypoglobus are the most frequently seen in the horizontal and vertical direction. These displacements are now provided as examples of globe position displacement (“The globe’s position may be displaced after trauma, which may affect the globe's position for instance, inward displacement (enophthalmos) or inferior displacement (hypoglobus)”).

Line 43: The sentence would be more grammatically correct without the word “causing”.

The word “causing” has been deleted.

Line 44: “Connective tissue” is a more appropriate term than “hard tissue”.

In this context, hard tissue is used for bony structures of the orbit. The terminology has been changed to “bony structures”.

Line 64: The word overview could be replaced with the term “visualization”.

This has been altered upon the reviewer’s suggestion.

  • Line 71: “Virtual surgical planning (VSP), intraoperative guidance, and postoperative evaluation.” — This list should follow a colon from the preceding sentence.

The colon has been added.

  • The authors should be clear about their purpose and intent of the paper.

The purpose of the manuscript was to demonstrate the current state of the art workflow for posttraumatic orbital reconstructions. To clarify this, the goal has been refrained in the introduction to:  “The main aim of this article is to provide a complete overview of the (CAS) workflow for orbital reconstruction, with in-depth description of the techniques embedded in the work-flow and with a special focus on treatment personalization through patient-specific implant design.”

  • The Introduction is long and contains a lot of generalized information that is readily available elsewhere. The manuscript would be easier to read and the focus of the paper (i.e. surgical approaches) would come across better if the Introduction was shortened.

The introduction has been shortened considerably. Since many readers of the Journal of Personalized Medicine may be from different specialties and not be familiar with orbital reconstructive surgery, the authors felt that additional background information might be necessary to improve readability for this broader audience.

Figure 1

  • This Figure does not add much to the paper and the authors should consider removing it to be more concise.

The figure has been replaced by an overview of the workflow as suggested by reviewer 1.

2.1 Diagnostics

  • Line 102: “enophthalmos, hypoglobus” should be replaced with a more general term, such as “globe displacement”, as these are not the only forms of displacement that are possible.

Enophthalmos and hypoglobus have been substituted by the more general term “globe displacement”.

  • Line 104: It should be specified that the Hertel mainly only measures relative *anterior-posterior* position. It does not necessarily measure displacement in other axes.

The text was altered to specify ventrodorsal globe position for the Hertel test.

Line 107: “There are no readily available reproducible measurement tools to measure hypoglobus” — This might be correct in terms of *clinical tools*, but the CT scan is able to accurately and reproducibly measure globe displacement.

A statement about this was already included in the manuscript, in the final sentence of the same paragraph: “Alternative methods have been proposed to quantify globe position differences based on imaging, but these have not been broadly implemented [22, 26].”

Line 108: The term for this technique is the “Hirschberg Test”.

The correct terminology, as suggested by the reviewer, has been included.

Line 115: “Impingement” is one of several possibilities. The EOMs can be severed and may also be entrapped by fractured bone.

Line 115: Any of the EOMs can be impinged and cause diplopia, so it is not advisable to specify this for only the “inferior ocular muscles”.

The text has been adjusted to include these possibilities: “Ocular motility can be disturbed by impingement or entrapment of the ocular muscles and surrounding soft tissue, but also by muscle edema, muscle injury, hemorrhage, emphysema, or motor nerve palsy.”

Line 119: It would be helpful to specify how soon the operations should take place in these cases.

The timing for orbital fractures with entrapment has been specified (“Absolute restrictions, as seen in trapdoor fractures, need to be treated shortly after the trauma.”). The reference is a systematic review on timing in orbital reconstruction that concludes that there is limited scientific evidence on timing in orbital reconstruction and a robust basis for guidelines is thus lacking. Indications and timing are closely related (“Some advocate a radical approach to prevent clinical symptoms [12], while others choose a more conservative approach with a delayed surgery if clinical symptoms develop [13]” line 57-59). In the response to reviewer 1, the authors explained why an elaborate discussion on these subjects was not included in the manuscript.

2.3 Virtual Surgical Planning

Line 169: It would be beneficial to list some indications for surgical reconstruction.

A few lines have been added to the introduction to address the indications for surgical reconstruction on a superficial level (“There is an.. and patient characteristics.” Line 53-57). As described in a previous answer to reviewer 1, the indication for surgical reconstruction is a controversial topic in facial trauma. The authors feel that the focus of the manuscript should lie on the 3D personalized workflow rather than indication or timing of reconstruction.

2.3.1 Patient-specific implant design

  • Line 201-203: “The prototype is imported into the virtual surgical planning and its fit evaluated; in contrast to the preformed implant setting, the position of the prototype is not adjusted, but its design.”— This sentence should be reformatted into 2 sentences (semi-colon should be replaced with a period). The second half of the sentence, after the semi-colon, does not read well and should be reworded.

The sentence has been split and the second part rephrased (“The prototype is imported into the virtual surgical planning and its fit evaluated. The position of the prototype is not adjusted in the virtual surgical planning to improve the fit, but the design of the prototype is adjusted and the novel prototype is reimported.”)

  • Line 235: It would be beneficial to elaborate on how the orbital volume is corrected. This is not always achieved by the implant alone.

The reviewer is correct that the orbital volume is not always corrected by the implant alone. This is also described in the manuscript (“by inserting additional spacers”).

  • Line 236: “anticipated” is more correct.

This was adjusted.

Table 1

This Table is a great idea to include, but it should be modified from its current form. The Table title should probably use the term “design considerations” instead of “design options”. The table should provide additional information in the Notes column (i.e. more than just the quotations from the referenced papers). There is no need for a reference column since it takes up too much space and the references can still be added elsewhere in the table without the full name of the author.

The reference column has been considerably shortened. “Design options” was changed to “design considerations”. The goal of the table was to illustrate the different design considerations and corresponding options as found in the literature. To achieve this, the references are used to tally what option was discussed and interesting quotes were listed in the “notes”  column. It is not clear which information should be incorporated in the “Notes” column, but the authors are open to suggestions from the reviewer. However, expanding the Table may affect readability.

Figure 6

This is a useful Figure. However, if there are labels (i.e. “A,B,C,D”), they should be described in the Figure legend.

The reviewer’s attentive comment has revealed a mismatch in the images and the subscripts of Figure 5 and 6. The images have been switched in the revised manuscript. The labels (A,B,C,D) are now described in the subscript of Figure 5.

  • The final sentence, “From top to bottom” does not make sense with the given image.

As described in the previous point, the images are now correctly paired with the subscripts.

2.4 Intraoperative Feedback

Line 294: “e.g. a specific spot on the implant” should be in parentheses.

The example was put in brackets.

2.5 Evaluation

Line 315: Is it correct to say that post-operative CT scans are *only* indicated when intra-operative CT scans were not performed or offered incomplete information?

An adjustment to this text was made to include other indications for postoperative CT scanning (“Post-operative CT-scans are indicated if intra-operative CT was not acquired (or offered incomplete information), or if indicated through clinical considerations in the follow-up period.”)

Line 328-329: “Interestingly, ocular motility can be improved by training the extraocular muscles to prevent and anticipate fibrosis.” — This is controversial as it is not widely established. The authors should include elaboration and references.

A reference has been added to substantiate this statement.

Round 2

Reviewer 1 Report

Thank you for the opportunity to judge this article again. The authors have invested time and effort in addressing the reviewers' comments and recommendations. One central point of criticism could be clarified - that this was an "Invited Article" on a defined question and not a classic review / systematic literature review was not apparent at first glance. Thus, from my point of view, all questions could be clarified, and nothing stands in the way of an acceptance of the article.